# ON PSEUDO-LABELING FOR CLASS-MISMATCH SEMI-SUPERVISED LEARNING

## ABSTRACT

Semi-Supervised Learning (SSL) methods have shown superior performance when unlabeled data are drawn from the same distribution with labeled data. Among them, Pseudo-Labeling (PL) is a simple and widely used method that creates pseudo-labels for unlabeled data according to predictions of the training model itself. However, when there are unlabeled Out-Of-Distribution (OOD) data from other classes, these methods suffer from severe performance degradation and even get worse than merely training on labeled data. In this paper, we empirically analyze PL in class-mismatched SSL. We aim to answer the following questions: (1) How do OOD data influence PL? (2) What are the better pseudo-labels for OOD data? First, we show that the major problem of PL is imbalanced pseudo-labels on OOD data. Second, we find that when labeled as their ground truths, OOD data are beneficial to classification performance on In-Distribution (ID) data. Based on the findings, we propose our model which consists of two components – Re-balanced Pseudo-Labeling (RPL) and Semantic Exploration Clustering (SEC). RPL re-balances pseudo-labels on ID classes to filter out OOD data while also addressing the imbalance problem. SEC uses balanced clustering on OOD data to create pseudo-labels on extra classes, simulating the process of training with their ground truths. Experiments show that our method achieves steady improvement over supervised baseline and state-of-the-art performance under all class mismatch ratios on different benchmarks.

## 1 INTRODUCTION

Deep Semi-Supervised Learning (SSL) methods are proposed to reduce dependency on massive labeled data by utilizing a number of cheap, accessible unlabeled data. Pseudo-Labeling (Lee et al., 2013) is a simple but effective and widely used method that creates pseudo-labels according to predictions of the training model itself. Then SSL can be transformed to standard supervised learning. Other representative SSL methods are consistency regularization (Laine & Aila, 2017; Tarvainen & Valpola, 2017; Miyato et al., 2019), holistic methods (Berthelot et al., 2019; Sohn et al., 2020) and generative methods (Kingma et al., 2014). The recent development of SSL shows that these methods have achieved competitive performance to supervised learning methods.

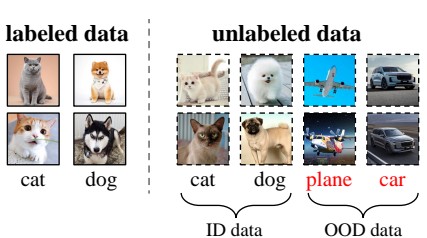

Figure 1: Realistic Semi-Supervised Learning may simultaneously contain unlabeled ID and OOD data. ID data come from the same classes as labeled data while OOD data come from classes that are not seen in labeled data.

However, all of these SSL methods achieve their good results based on an assumption that the unlabeled data are drawn from the same distribution as the labeled data. This assumption can be easily violated in real-world applications. One of the common cases is that some unlabeled data come from unseen classes. For example, as is illustrated in Figure 1, in image classification, we can collect a lot of unlabeled images from the internet but usually they cover broader category concepts than labeled data. Oliver et al. (2018) have shown that on such class-mismatched conditions, performance of traditional SSL methods is damaged. To deal with this problem, several methods have been proposed. These methods include filtering out OOD data (Yu et al., 2020; Chen et al., 2020), down weighting

OOD data (Chen et al., 2020) and re-use OOD data by neural style transfer (Luo et al., 2021) or self-supervised learning (Huang et al., 2021). Although these methods achieve good results, why do OOD data damage performance and how will OOD data help remain unclear. Here, we focus on analyzing Pseudo-Labeling (PL) in class-mismatched SSL and give some answers to these two questions.

In this paper, we empirically analyze PL in class-mismatched SSL setting. These experiments aim to answer the following questions: (1) How do OOD data influence PL? (2) What are the better pseudo-labels for OOD data? For question (1), we investigate pseudo-labels created by PL. The main finding is that pseudo-labels on OOD data tend to be imbalanced while on ID data, it remains balanced. We further show that PL's performance is damaged due to such imbalance on OOD data. For question (2), several strategies for labeling OOD data are investigated. We conclude that it is beneficial when labeling OOD data as a class different from ID data, and the performance can be further improved when the pseudo-labels partition unlabeled OOD data into their semantic clusters.

Based on the experimental analysis, we propose a two-branched model called $\Upsilon$-Model, which processes unlabeled data according to their confidence score on ID classes. The first branch performs re-balanced pseudo-labeling on high-confidence data. It utilizes the property of imbalanced pseudo-labels on OOD data, truncating the number of pseudo-labeled data for each class to their minimum. This procedure filters out many OOD data and also prevents the negative effect of imbalanced pseudo-labels. For the other branch, semantic exploration clustering is performed on low-confidence data. They are considered as OOD data and their semantics will be mined by clustering into different partitions on extra classes. The clustering result provides better pseudo-labels for these OOD data than vanilla PL. Experiments on different SSL benchmarks show that our model can achieve steady improvement in comparison to supervised baseline. We summarize our contributions as follows:

- We analyze the Pseudo-Labeling model for ID and OOD data. The findings lead to two primary conclusions: (1) Imbalance of pseudo-labels on OOD data damages PL's performance. (2) Best pseudo-labels for unlabeled OOD data are those different from ID classes and partitioning them into their semantic clusters.

- We propose our two-branched $\Upsilon$-Model. One branch re-balances pseudo-labels on ID classes and filter out OOD data. The other branch explores semantics of OOD data by clustering on extra classes.

- Experiments on different SSL benchmarks empirically validate effectiveness of our model.

## 2 PRELIMINARY

### 2.1 CLASS-MISMATCHED SSL

Similar to the SSL problem, the training dataset of the class-mismatched SSL problem contrains $n$ ID labeled samples $\mathcal{D}_l = \{(\mathbf{x}_{li}, y_{li})\}_{i=1}^n$ and $m$ unlabeled samples $\mathcal{D}_u = \{\mathbf{x}_{ui}\}_{i=1}^m$, (usually, $m \gg n$,) $y_{li} \in \mathcal{Y}_{ID} = \{1, \ldots, K_{ID}\}$, while different from SSL, the underlying ground truth $\mathbf{y}_u$ of unlabeled data may be different from labeled data. *i.e.*, $y_{uj} \in \mathcal{Y}_{ID} \cup \mathcal{Y}_{OOD}, \mathcal{Y}_{OOD} = \{K_{ID} + 1, \ldots, K_{ID} + K_{OOD}\}$. The goal of class-mismatched SSL is to **correctly classify ID samples into** $\mathcal{Y}_{ID}$ using labeled set with ID samples and unlabeled set possibly with OOD samples.

### 2.2 PSEUDO-LABELING

*Pseudo-Labeling* (PL) leverages the idea that we can use the model itself to obtain artificial labels for unlabeled data (Lee et al., 2013). PL first perform supervised learning on labeled data to get a pre-trained model $f$, which outputs the probability of belonging to each ID class. It then creates the pseudo-labels for each unlabeled sample:

$$y' = \begin{cases} \arg\max_{y \in \mathcal{Y}_{ID}} f(y|\mathbf{x}) & , \quad c(\mathbf{x}) > \tau \\ \text{reject} & , \quad \text{otherwise} \end{cases}, \tag{1}$$

$$c(\mathbf{x}) = \max_{y \in \mathcal{Y}_{ID}} f(y|\mathbf{x}), \tag{2}$$

where $c(\mathbf{x})$ is the confidence score for $\mathbf{x}$. All the pseudo-labeled unlabel data will be treated as labeled data for the next supervised learning generation. PL iteratively performs supervised learning and pseudo-label creation until stop condition.

# 3    ANALYSIS OF PSEUDO-LABELING IN CLASS-MISMATCHED SSL

In class-mismatched SSL, vanilla PL can only create pseudo-labels on ID classes even for OOD data. We will analyze how these OOD data influence vanilla PL and what are the better pseudo-labels for them in this section.

## 3.1    SETUP

We use CIFAR-10 (Krizhevsky et al., 2009) as our experimental dataset. The data set contains 10 categories – 6 animal classes and 4 vehicle classes. Following Guo et al. (2020), we perform a classification task on animal classes (denoted as class 0-5) and select 400 images per class to construct the labeled data set, *i.e.*, 2,400 labeled examples. The other 4 vehicle classes are taken as OOD classes (denoted as classes 6-9). 20,000 images are randomly selected from all the 10 classes as the unlabeled data set. We vary the ratio of unlabeled images to modulate class distribution mismatch. For example, the extent is 50% means half of the unlabeled data comes from animal classes and the others come from vehicle classes. We use Wide-ResNet-28-2 (Zagoruyko & Komodakis, 2016) as our backbone. We also adopt data augmentation techniques including random resized crop, random color distortion and random horizontal flip. We train our network for 400 epochs. For each epoch, we iterate over the unlabeled set and random sample labeled data, each unlabeled and labeled mini-batch contains 128 samples. We adopt Adam as the optimization algorithm with the initial learning rate $3 \times 10^{-3}$. We report the averaged accuracy of the last 20 epochs, pretending there is no reliable (too small) validation set to perform early stop (Oliver et al., 2018).

## 3.2    IMBALANCE OF PSEUDO-LABELS ON OOD DATA

In this section, we analyze the **pre-trained model** that creates the first set of pseudo-labels, and the **final model** trained by Pseudo-Labeling.

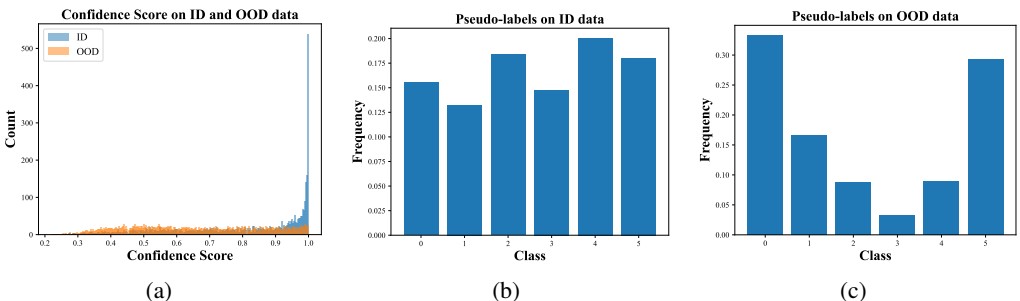

|     (a)     |     (b)     |     (c)     |

Figure 2: Analysis of pre-trained model. (a) Histogram of the computed confidence scores $c(x)$ over ID and OOD data. (b) On ID data, pseudo-label distribution is balanced since they share the same distribution with the labeled data. (c) On OOD data, the pseudo-label distribution is imbalanced.

**Pretrained model.**   First, we draw the distribution of confidence score on OOD data and ID data. Figure 2(a) tells that, like what is concluded in OOD detection (Hendrycks & Gimpel, 2017), proportion of high-confidence data in ID data is larger than OOD data. However, in class-mismatched SSL, the unlabeled data are in much larger quantities. When the class mismatch ratio is large, there are quite a few OOD data with high confidence scores. We will show in the final model experiments that these high-confidence OOD data damages performance. Secondly, we study pseudo-labels on both ID data and OOD data. Figure 2(b) shows that pseudo-labels ID data is balanced. However, they are rather imbalanced on OOD data (Figure 2(c)). This is attributed to the different distribution they are drawn from. Samples with certain pattern bias to certain classes. ID data bias to ID classes uniformly because they are sampled by the same distribution. However, with little probability, OOD data will also bias to ID classes uniformly since they have little relevance to ID data.

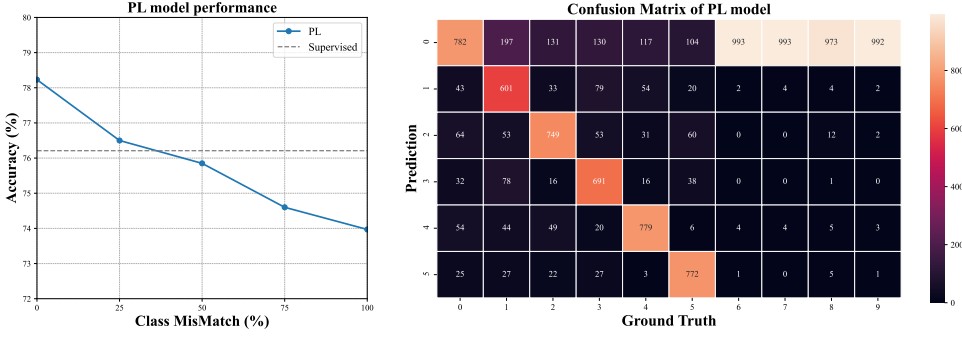

(a) Performance of PL  (b) Confusion Matrix of PL model (Mismatch Ratio = 100%)

Figure 3: (a) PL model degrades as the mismatch ratio increases. (b) Confusion matrix of PL model when the mismatch ratio = 100%. It demonstrates that the imbalance of pseudo labels on OOD data affects PL's performance. A lot of ID samples with class 1-5 are misclassified into class 0. Also, as the PL process goes on, the imbalance pseudo-labels on OOD data get even worse.

**Final Pseudo-Labeling model.** As an old saying goes, a good beginning is half done. However, such imbalance of the first set of pseudo-labels starts PL model badly when there is a large portion of OOD data, putting the model in danger of imbalanced learning. We run vanilla PL and show that the *imbalance of pseudo-labels harms the performance.* Figure 3(a) shows the performance of PL model with different OOD ratios. In accord with Oliver et al. (2018), PL model degrades as the portion of OOD data gets larger. Figure 3(b) displays the confusion matrix of the PL model on the whole test set containing both ID and OOD data. Since only 6 classes are known to us, the confusion matrix is a rectangle. We can see almost all the OOD samples (class 6-9) are classified as class 0, which means the imbalance effect on OOD data gets even worse as the PL training goes on. The possible reason is that, unlike Pseudo-labeling on ID data, supervision of labeled data can not help correct pseudo-labels on OOD data. Thus the imbalance continuously deteriorates. The imbalance on OOD data also influences classification performance on ID data. Samples of major classes (class 0) overwhelm the loss and gradient, leading to a degenerate model (Lin et al., 2017). We can see the PL model mistakenly classifies many of data with class 1-5 into class 0.

### 3.3 Pseudo-Labeling Strategy for OOD data

The previous section shows OOD data hurt performance of vanilla PL. Then here comes the question: Assuming that we already know which data are OOD, **how do we use these OOD data? Is omitting them the best way? If not, what are the better pseudo-labels for them?** To answer these questions, we investigate four strategies to create pseudo-labels for OOD data:

- **Baseline.** This baseline omits all the OOD data and only trains on the labeled ID data.
- **Re-Assigned Labeling.** This strategy assigns data of each OOD class to an ID class. It ensures that different OOD class is assigned to different ID class, keeping the semantics unchanged between OOD classes. For example, (ship, trunk, airline, automobile) can be assigned to (bird, cat, deer, dog). This strategy can be seen as training a classifier of "super-classes".
- **Open-Set Labeling.** This strategy is named after the related setting – Open-Set Recognition (Scheirer et al., 2013; Bendale & Boult, 2016). This strategy treats all OOD data as one unified class $K_{ID} + 1$. Thus this model outputs probability over $K_{ID} + 1$ classes.
- **Oracle Labeling.** This strategy uses the ground truth of OOD data. Thus this model outputs probability over $K_{ID} + K_{OOD}$ classes.

Note that Open-Set Labeling and Oracle Labeling can classify samples into more than $K_{ID}$ classes. However, during evaluation, we only classify samples into $K_{ID}$ ID classes. For these model, the predicted label $\hat{y}$ of a test sample $\mathbf{x}$ is calculated as:

$$\hat{y}(x) = \arg\max_{y \in \mathcal{Y}_{ID}} f(y|\mathbf{x}) \tag{3}$$

The overall comparison of the four strategies is illustrated in Figure 4. We also report test accuracy when class mismatch ratio is 100%. From Figure 4, we can get several important conclusions. (1)

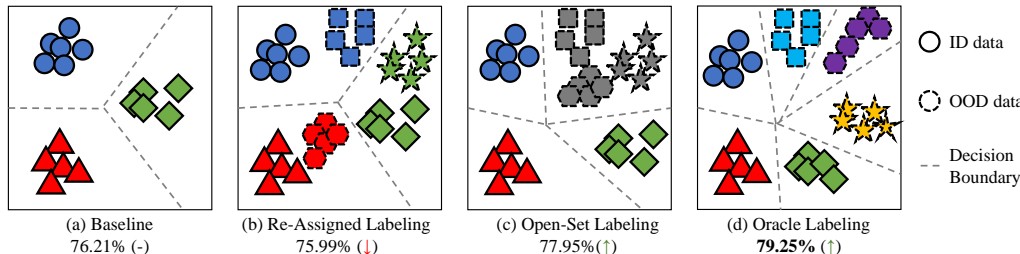

Figure 4: Four strategies of how to label OOD data. Different shapes represent different ground truths. Data with the same color are labeled as the same classes. A shape with a solid outline means it is an ID sample while OOD data are represented with dashed line. (a) No Labeling acts as a baseline where OOD data are omitted. (b) Re-Assigned Labeling re-labels OOD data to certain ID classes. (c) Open-Set Labeling labels all the OOD data as a unified class. (d) Oracle Labeling uses the ground truths of OOD data. Test accuracy under $100\%$ class mismatch ratio is reported.

Re-Assigned Labeling underperforms baseline a little[1]. This indicates that assign samples with OOD classes to ID classes does not help the model distinguish between ID classes even if we somehow know which OOD data are semantically different. It also reveals that performing vanilla PL on OOD data may never help even if we do it perfectly. (2) Open-Set Labeling outperforms baseline, which indicates it improves the performance if we label the OOD data as a class other than ID classes. (3) We can see Oracle Labeling improves the performance and achieves the best result among the four strategies. It means that in addition to label OOD data as extra classes, if we can further assign OOD data with different semantics to different classes, the model will achieve better results.

## 3.4 Summary of Section

In this section, we study the behavior of Pseudo-Labeling model in class-mismatched SSL. We summarize several important conclusions here:

Conclusion 1: Classification model trained with labeled ID data creates imbalanced pseudo-labels on OOD data while on ID data, it remains balanced.

Conclusion 2: The vanilla PL process makes the imbalance of pseudo-labels deteriorate, damaging the classification performance on ID data.

Conclusion 3: Labeling OOD data as ID classes does not help and may even perform a little worse.

Conclusion 4: It is beneficial to label OOD data as extra classes different from ID classes. If we can further label semantically different OOD data as different classes, the performance can be further improved.

## 4 Method

Based on the findings in Section 3, we proposed $\Upsilon$-Model (named after its shape) for class-mismatched SSL. $\Upsilon$-Model trains a classifier $f$ that will output the posterior distribution over $K_{ID} + K$ classes, *i.e.*, $f(\mathbf{y}|\mathbf{x}) \in \mathbb{R}^{K_{ID}+K}, \mathbf{1}^\top f(\mathbf{y}|\mathbf{x}) = 1$. $K$ is the number of extra classes, which can be known in advance (*i.e.*, $K = K_{OOD}$) or be set as a hyper-parameter. Similar to vanilla PL, we define confidence with the same form as Equation 2. However, this confidence is a little different from its original definition in Hendrycks & Gimpel (2017), for we only calculate the maximum probability of the $K_{ID}$ classes instead of all. Therefore, we rename it to **In-Distribution confidence (ID confidence)**. For evaluation, we predict labels using Equation 3. $\Upsilon$-Model aims to solve the following questions:

Problem 1: how to avoid imbalanced pseudo-labels in PL model? (Conclusion 1, 2)

Problem 2: how to avoid labeling OOD data as ID? (Conclusion 3)

Problem 3: how to create proper pseudo-labels for unlabeled OOD data? (Conclusion 4)

---

[1]Assign 4 OOD classes to 4 ID classes of 6 causes imbalance. But we test the accuracy on selected 4 classes and find they show a similar result.

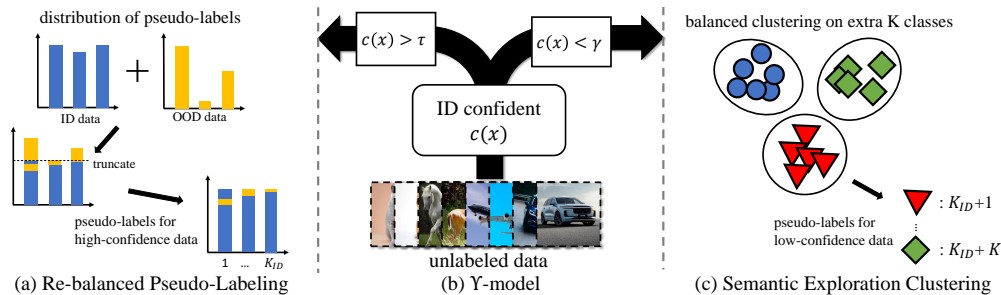

Figure 5: Illustration of $\Upsilon$-Model and its two main branches. (b) is the main structure of $\Upsilon$-Model where we judge by the ID confidence if certain unlabeled data belongs to ID classes or not. The data with high confidence will perform Re-balanced Pseudo-Labeling, while those with low confidence will get their pseudo-labels by Semantic Exploration Clustering. (a) Re-balanced Pseudo-Labeling truncates the number of pseudo-labeled data to the minimum, making the pseudo-labels balanced and filtering out OOD data. (c) Semantic Exploration Clustering simulates the process of learning from ground truth labels of OOD data, creating pseudo-labels on extra $K$ classes by clustering.

$\Upsilon$-Model consists of two main branches – Re-balanced Pseudo-Labeling (RPL) and Semantic Exploration Clustering (SEC). RPL acts on high-confidence data to solve Problem 1, 2. SEC acts on low-confidence data to solve Problem 3. We describe the two branches in the following sections. The overview of $\Upsilon$-Model is illustrated in Figure 5.

## 4.1 RE-BALANCED PSEUDO-LABELING

As is illustrated in Section 3.3, the main problem of vanilla PL is that a large number of OOD data with high confidence scores have imbalanced pseudo-labels. One possible solution is re-weighting the unlabeled sample (Guo et al., 2020) or using other methods in the imbalance learning field. However, even if we solve the problem of imbalance learning, labeling OOD data as ID classes also may damage the performance (Conclusion 3). In this paper, we use a simple method – Re-balanced Pseudo Labeling – to simultaneously solve imbalance (Problem 1) and incorrect recognition (Problem 2). It produces a set $\mathcal{P}$ of pseudo-labeled samples in three steps:

$$N = \min_{y \in \mathcal{Y}_{ID}} |\{\mathbf{x} \in \mathcal{D}_u \mid f(y \mid \mathbf{x}) > \tau\}|, \tag{4}$$

$$\tau_y = Nth\_biggest(\{f(y \mid \mathbf{x}) \mid \mathbf{x} \in \mathcal{D}_u\}), \qquad y = 1, 2, \ldots, K_{ID}, \tag{5}$$

$$\mathcal{P} = \bigcup_{y \in \mathcal{Y}_{ID}} \{(\mathbf{x}, y) \mid f(y \mid \mathbf{x}) \geq \tau_y, \mathbf{x} \in \mathcal{D}_u\}, \tag{6}$$

where $Nth\_biggest$ represents the $N$–th biggest value of the given set. RPL first calculates the minimum number of pseudo-labeled samples for each ID class by Equation 4. Then it truncates the number of pseudo-labels of each ID class to that number by Equation 5, 6. The process of RPL is illustrated in Figure 5(a). First, it enforces the pseudo labels on ID classes to be balanced, solving Problem 1. Second, as is shown in Section 3, the set of high-confidence data is a mixture of ID and ODD data. Due to Conclusion 1, the pseudo-label distribution of such a set is a sum of imbalanced and balanced ones, thus still imbalanced. However, by selecting only top-$N$ confident samples for each ID class, we will keep ID data and omit many OOD data since confidence on ID data tends to be higher than OOD data (Hendrycks & Gimpel, 2017). This process solves Problem 2.

## 4.2 SEMANTIC EXPLORATION CLUSTERING

As is demonstrated in Section 3.3, if we know a set of samples is OOD, it will improve the performance if we label them as a unified class $K_{ID} + 1$. But the best way is to use their ground truths (Conclusion 4). However, it is impossible to access their ground truths since they are unlabeled. We resort to using Deep Clustering methods (Caron et al., 2018; Asano et al., 2020) to mine their semantics and approximate the process of learning with the ground truths. Here, we use the balanced clustering method in Asano et al. (2020); Caron et al. (2020) to create pseudo-labels for these OOD data. Assuming there are $M$ samples recognized as OOD, we first compute their soft targets:

$$\min_{Q \in U(K,M)} \langle Q, -\log P \rangle, \quad U(K,M) := \left\{ Q \in \mathbb{R}_+^{K \times M} \mid Q\mathbf{1} = \frac{1}{K}\mathbf{1}, Q^{\top}\mathbf{1} = \frac{1}{M}\mathbf{1} \right\}, \tag{7}$$

where $P \in \mathbb{R}_+^{K \times M}, P_{ij} = \hat{f}(K_{ID} + i|\mathbf{x}_j)$. $\hat{f}$ is the normalized posterior distribution on extra classes, *i.e.*, $\hat{f}(K_{ID}+i|\mathbf{x}_j) = f(K_{ID}+i|\mathbf{x}_j)/\sum_{k=1}^{K} f(K_{ID}+k|\mathbf{x}_j)$. We use *the Sinkhorn-Knopp algorithm* Cuturi (2013) to optimize $Q$. Once we get $Q$, we harden the label by picking the class with the maximum predicted probability and mapping it to the extra $K$ classes:

$$\hat{y}_j = K_{ID} + \arg\max_i Q_{ij}. \tag{8}$$

$\hat{y}_j$ is used as the pseudo-label for $\mathbf{x}_j$. We perform SEC on the set of data with ID confidence lower than a threshold $\gamma$, *i.e.*, $\{\mathbf{x}|c(x) < \gamma\}$.

## 5 RELATED WORK

**Class-Mismatched Semi-Supervised Learning.** Deep Semi-Supervised Learning suffers from performance degradation when there are unseen classes in unlabeled data (Oliver et al., 2018). As the proportion of such out-of-distribution (OOD) data get larger, the performance drop more. To cope with such a class-mismatch problem, several methods are proposed. Chen et al. (2020) formulate a sequence of ensemble models aggregated accumulatively on-the-fly for joint self-distillation and OOD filtering. Guo et al. (2020) re-weight the unlabeled data by meta-learning to decrease the negative effect of OOD data. Huang et al. (2020) recycle transferable OOD data by means of adversarial learning. Different from all these methods, we conduct a comprehensive study on Pseudo-Labeling (PL) and give useful guidance on how to do better in class-mismatched SSL.

**Pseudo-Labeling.** The method of Pseudo-Labeling, also known as self-training, is a simple and effective way for Deep Semi-Supervised Learning (Lee et al., 2013; Shi et al., 2018; Arazo et al., 2020; Iscen et al., 2019). Despite its simplicity, it has been widely applied to diverse fields such as image classification (Xie et al., 2020), natural language processing (He et al., 2020) and object detection (Rosenberg et al., 2005). The use of a hard label makes Pseudo-Labeling closely related to entropy minimization (Grandvalet & Bengio, 2004).

**Deep Clustering.** Deep clustering methods improve ability of traditional cluster methods by leveraging the representation power of DNNs. A common means is to transform data into low-dimensional feature vectors and apply traditional clustering methods (Yang et al., 2017; Caron et al., 2018). In Self-Supervised Learning, clustering methods are used to learning meaningful representation for downstream tasks (Caron et al., 2018; Asano et al., 2020; Caron et al., 2020). Modern Deep Clustering can learn semantically meaningful clusters and achieves competitive results against supervised learning (Gansbeke et al., 2020).

## 6 EXPERIMENTS

To validate the effectiveness of our $\Upsilon$-Model, we conduct experiments on different benchmarks.

**Dataset.** We test our methods on two datasets as in Oliver et al. (2018): (1) **CIFAR10**: we use the same configuration as in Section 3.1. **SVHN**: The data set contains 10 categories – digits "0"-"9". We select the first "0"-"5" as ID classes and the rest as OOD. For each class, we randomly select 100 images as labeled data. Meanwhile, 20,000 images are randomly selected from all the 10 classes as the unlabeled data set. The class-mismatch ratio is set as $\{0\%, 25\%, 50\%, 75\%, 100\%\}$.

**Implementation Details.** We use the same network and training protocol as Section 3.1. We first train a classification model only on labeled data for 100 epochs without RPL and SEC. We update pseudo-labels every 2 epochs. For both datasets, we set $\tau = 0.95$, $\gamma = 0.3, K = 4$. We use an exponential moving average model for final evaluation as in Athiwaratkun et al. (2019).

### 6.1 COMPARE WITH TRADITIONAL SSL METHODS

In this subsection, we compare our methods with four traditional SSL methods – Pseudo-Labeling (Lee et al., 2013), $\Pi$-Model (Laine & Aila, 2017), Mean Teacher (Tarvainen & Valpola, 2017) and VAT (Miyato et al., 2019). Figures 6(a), 6(b) show the results. Traditional methods suffer from performance degradation as the mismatch ratio increases. They usually get worse than the supervised baseline when the mismatch ratio is larger than $50\%$ on CIFAR10 and SVHN. In contrast, our methods get steady improvement under all class mismatch ratios. The reasons can be attributed as follows. First, our method is aware of the existence of OOD data. We do not treat OOD data

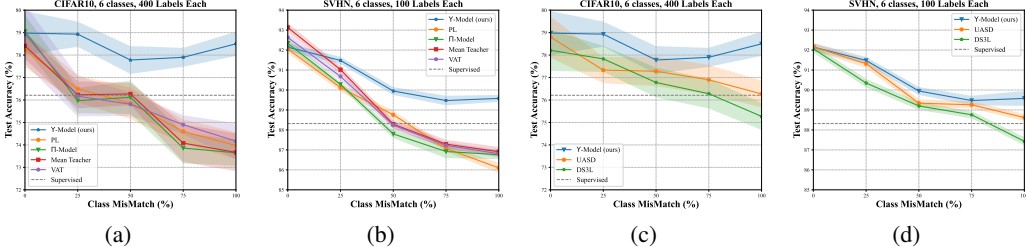

(a)        (b)        (c)        (d)

Figure 6: Comparison with existing methods on CIFAR10 and SVHN dataset with Wide-ResNet-28-2 network. Class mismatch ratios are varied. The shaded regions with the curves indicate the standard deviations of the accuracies over five runs. (a) (b) Comparison with traditional SSL methods. These methods suffer from performance degradation as the mismatch ratio increases. (c) (d) Comparison to two existing class-mismatched SSL methods – UASD and DS$^3$L. Our methods perform better in almost all the experimental setups.

like ID data, which can hurt performance. Second, we reuse OOD data by exploring their semantics which proves to be useful in Section 3.3. Therefore, even when the class-mismatch ratio gets $100\%$, performance of $\Upsilon$-Model is still better than supervised baseline.

## 6.2 COMPARE WITH CLASS-MISMATCHED SSL METHODS

In this subsection, we compare our method with two existing class-mismatched SSL methods – UASD (Chen et al., 2020) and DS$^3$L (Guo et al., 2020). For a fair comparison, we use Pseudo-Labeling as the base method of DS$^3$L. From Figures 6(c), 6(d), we can see our methods are superior to these two methods in all settings. It is noticeable that DS$^3$L underperforms supervised baseline when all the unlabeled data are drawn from OOD classes. This is attributed to the fact that DS$^3$L uses a down weighting strategy to alleviate the negative effect of OOD data and does not change the form of unsupervised loss. But we have shown in Section 3.3 that labeling OOD data as ID classes damage performance anyhow. On the contrary, $\Upsilon$-Model uses the OOD data in the right way – simulating the process of training them with their ground truth labels. As a result, our method shows superiority especially under a large class-mismatch ratio. We also notice that the performance curve of $\Upsilon$-Model appears a U-shape (obvious on CIFAR10). A possible reason is that RPL and SEC compete with each other. RPL tends to make samples get a high prediction on ID classes while SEC tends to make samples get a high prediction on OOD classes. When the class-mismatch ratio reaches $0\%$ ($100\%$), RPL (SEC) dominants the other. In this circumstance, one works without any disturbance of the other. However, when the class-mismatch ratio is $50\%$, they compete fiercely with each other, causing many incorrectly recognized ID or OOD samples.

## 6.3 ABLATION STUDY

In this section, we validate the functionality of RPL and SEC. We conduct experiments on CIFAR10 benchmark as in the analysis section 3.

**Validation of effectiveness of RPL and SEC.** We conduct ablation studies under different class-mismatch ratios and report the averaged test accuracy and standard deviation of five runs. As usual, we vary the class-mismatch ratio. Table 1 displays the results. Firstly, comparing the first line and second line of the table, RPL not only outperforms vanilla PL in high class-mismatch ratio scenario but also improves in low class-mismatch ratio scenario. This reveals that balanced pseudo-labels always help since once the model creates imbalanced pseudo-labels, it will deteriorate when there are not enough measures to correct it. Secondly, comparing the second and third line, it shows that RPL alone alleviate the performance degradation but it can not prevent it, in accord with Conclusion 3. When using SEC, $\Upsilon$-Model gets better results than supervised baseline when the class-mismatch ratio is high. Besides, comparing the third and last lines, we see that when cluster OOD data and exploring their semantics instead of using a unified class to label them, the performance improves.

**RPL helps filter out OOD data and solve imbalance problem.** It is noticeable that the last two lines of Table 1 show that without RPL, SEC alone can not achieve better performance than supervised baseline. We will show the reason here. Figure 7(a) plots the proportion of OOD data that are pseudo-labeled as ID classes. It reveals that without RPL, *i.e.*, using vanilla PL, the number

Table 1: Validation of RPL and SEC. The experiments are conducted on CIFAR10 with Wide-ResNet-28-2 backbone. Class-mismatch ratio varies from 0 to $100\%$. Check RPL or not means using RPL or vanilla PL. $K = 0$ means we do not use SEC. $K = 1$ means we label all the low confidence data as a unified class $K_{ID} + 1$.

| RPL | SEC | $K$ | Class-Mismatch Ratio (%) | | | | |
|---|---|---|---|---|---|---|---|
| | | | 0 | 25 | 50 | 75 | 100 |
| | | 0 | $78.23 \pm 0.39$ | $76.50 \pm 0.30$ | $75.85 \pm 0.35$ | $74.60 \pm 0.27$ | $73.97 \pm 0.30$ |
| ✓ | | 0 | $\mathbf{78.76 \pm 0.49}$ | $76.92 \pm 0.32$ | $76.80 \pm 0.31$ | $75.48 \pm 0.27$ | $75.38 \pm 0.27$ |
| ✓ | ✓ | 1 | $\mathbf{78.86 \pm 0.37}$ | $77.17 \pm 0.55$ | $77.12 \pm 0.37$ | $76.88 \pm 0.35$ | $77.27 \pm 0.39$ |
| | ✓ | 4 | $78.11 \pm 0.45$ | $77.43 \pm 0.29$ | $77.46 \pm 0.37$ | $76.38 \pm 0.37$ | $74.18 \pm 0.32$ |
| ✓ | ✓ | 4 | $\mathbf{78.98 \pm 0.49}$ | $\mathbf{78.93 \pm 0.28}$ | $\mathbf{77.78 \pm 0.31}$ | $\mathbf{77.90 \pm 0.21}$ | $\mathbf{78.50 \pm 0.27}$ |
| baseline | | | | | $76.21 \pm 0.21$ | | |

of incorrectly recognized OOD data keep increasing as the training proceed. While with RPL, this ratio rapidly drops to 0. This proves that RPL help filter out OOD data by utilizing the imbalance property of OOD data. Further, we present the confusion matrix of $\Upsilon$-Model on the full test set (all the 10 classes) of CIFAR10. Compared to vanilla PL in Figure 3(b), $\Upsilon$-Model does not suffer from imbalance problem, as a result of which, its performance is not degraded.

**Effect of extra class number $K$.** We vary the number of extra classes $K$. Figure 7(c) shows the result on CIFAR10 with class mismatch ratio $100\%$. The gray dashed line is the supervised baseline. The red dashed line is the Oracle Labeling strategy in Section 3.3. It is the upper bound of $\Upsilon$-Model in this setup. Without SEC ($K = 0$), $\Upsilon$-Model underperform supervised baseline. Using SEC ($K \geq 1$), $\Upsilon$-Model is always better than baseline. Also, it reaches its best performance when $K$ equals the actual number of OOD classes. This demonstrates that by simulating the process of training on OOD data with ground truth, SEC helps classification model on ID data.

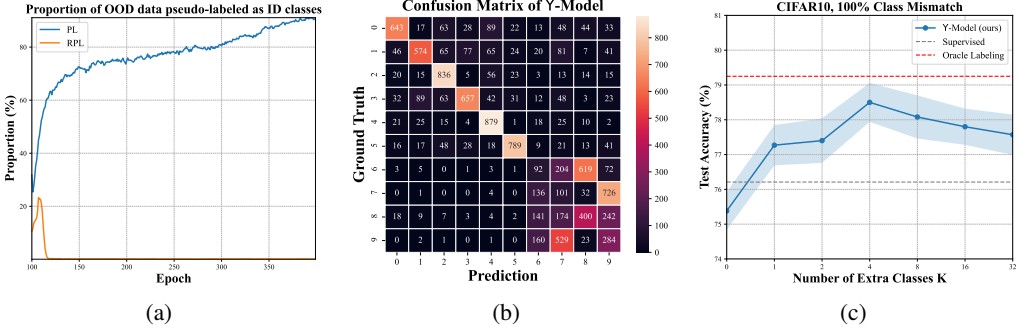

|          (a)          |          (b)          |          (c)          |

Figure 7: Three experiments on CIFAR10 benchmark to validate the functionality of RPL and SEC. (a) The proportion of OOD data pseudo-labeled as ID classes with the training goes on. Vanilla PL makes this ratio keep increasing while RPL makes it drop to 0. It demonstrates that RPL help filter out OOD data. (b) Confusion matrix on the full test set of CIFAR10. RPL solves the imbalance problem. (c) Test accuracy versus the number of extra classes $K$. It reaches the best performance when $K$ equals the actual number of OOD classes.

## 7 CONCLUSION

In this paper, we analyze Pseudo-Labeling in class-mismatched semi-supervised learning where there are unlabeled OOD data from other classes. We show that Pseudo-Labeling suffers from performance degradation due to imbalanced pseudo-labels on OOD data. The correct way to use OOD data is to label them as classes different from ID classes while also partitioning them according to their semantics. Based on the analysis, we proposed $\Upsilon$-Model and empirically validate its effectiveness. We believe our findings are not only beneficial to PL methods, but also inspiring to other methods like consistency regularization and holistic methods on how to effectively use OOD data.

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

## A    ALGORITHM

---

**Algorithm 1** $\Upsilon$-Model algorithm

---

**Input:** Labeled dataset $\mathcal{D}_l = \{(\mathbf{x}_{li}, y_{li})\}_{i=1}^n$, and unlabeled dataset $\mathcal{D}_u = \{\mathbf{x}_{ui}\}_{i=1}^m$; Classification model $f_\phi$ parameterized with $\phi$, ID class number $K_{ID}$, extra class number $K$, total epoch number $E$, pretrain epochs $E_{pt}$, interval to update pseudo-labels $E_{pl}$, pseudo-labeled set $\mathcal{P}$, confidence calculation function $c$.

1:  **function** REBALANCEDPSEUDOLABELING($\mathcal{D}, f, \tau$)
2:      $N \leftarrow \min_{y \in \mathcal{Y}_{ID}} |\{\mathbf{x} \in \mathcal{D}_u \mid f(y \mid \mathbf{x}) > \tau\}|$
3:      $\tau_y \leftarrow Nth\_biggest(\{f(y \mid \mathbf{x}) \mid \mathbf{x} \in \mathcal{D}_u\}), \quad y = 1, 2, \ldots, K_{ID}$
4:      $\mathcal{P} \leftarrow \bigcup_{y \in \mathcal{Y}_{ID}} \{(\mathbf{x}, y) \mid f(y \mid \mathbf{x}) \geq \tau_y, \mathbf{x} \in \mathcal{D}_u\}$
5:      **return** $\mathcal{P}$
6:  **function** SEMANTICEXPLORATIONCLUSTERING($\mathcal{D}, f, \gamma$)
7:      $S \leftarrow \{\mathbf{x} | c(x) < \gamma\}$
8:      $M \leftarrow |S|$
9:      $P_{ij} \leftarrow f(K_{ID} + i | \mathbf{x}_j) / \sum_{k=1}^K f(K_{ID} + k | \mathbf{x}_j), \quad i = 1, 2, \ldots, K, \quad j = 1, 2, \ldots, M$
10:     Solve 7 by Sinkhorn-Knopp algorithm and get $Q$
11:     $\hat{y}_j \leftarrow K_{ID} + \arg\max_i Q_{ij}, \quad j = 1, 2, \ldots, M$
12:     $\mathcal{C} \leftarrow \{(\mathbf{x}_j, \hat{y}_j)\}_{j=1}^M$
13:     **return** $\mathcal{C}$
14: **for** e = 1 to $E$ **do**
15:     **if** e $< E_{pt}$ **then**
16:         train $f_\phi$ with standard supervised learning on $\mathcal{D}_l$                    ▷ Pre-training phase
17:         L =
18:     **else**
19:         train $f_\phi$ with standard supervised learning on $\mathcal{D}_l \cup \mathcal{P}$      ▷ PL training phase
20:     **if** e $\leq E_{pt}$ **and** e $\% E_{pl} = 0$ **then**
21:         $\mathcal{P} \leftarrow \emptyset$
22:         $\mathcal{P}_\tau \leftarrow$ REBALANCEDPSEUDOLABELING($D_u, f_\phi, \tau$)              ▷ Perform RPL
23:         $\mathcal{P}_\gamma \leftarrow$ SEMANTICEXPLORATIONCLUSTERING($D_u, f_\phi, \gamma$)    ▷ Perform SEC
24:         $\mathcal{P} \leftarrow \mathcal{P}_\tau \cup \mathcal{P}_\gamma$
25: **return** classification model $f_\phi$

---

## B    EMBEDDING VISUALIZATION

We visualize the embedding of supervised baseline, vanilla PL and $\upsilon$-Model on CIFAR10's test set with all the classes by t-SNE. Figure 8 shows the result. OOD data are mixed with ID data since the supervised baseline does not see unlabeled OOD data. PL mixes OOD data with samples of certain classes (class 0). This is attributed to that their pseudo-labels are biased toward this class. Also, we can not clearly distinguish between OOD classes. In contrast, $\Upsilon$-Model can not only make ID data distinguishable but also forms meaningful clusters on OOD data.

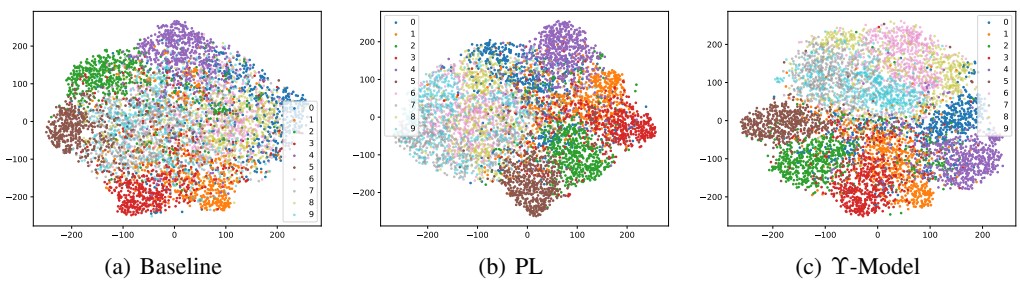

(a) Baseline                    (b) PL                    (c) $\Upsilon$-Model

Figure 8: t-SNE visualization of supervised baseline, vanilla PL and $\Upsilon$-Model on CIFAR10's test set. Class 0-5 are ID classes, shown by non-transparent circles. Class 6-9 are OOD classes, represented by semi-transparent circles.

## C  UNIVERSALITY OF ANALYSIS CONCLUSIONS

To illustrate the universality of the conclusion in Section 3. We experiment on totally five kinds of datasets. ( We use $(n/m)$ to represent $n$ ID classes and $m$ OOD classes.)

- **CIFAR10 (6/4)** and **CIFAR10 (5/5)** : both created from **CIFAR10** (Krizhevsky et al., 2009).**CIFAR10 (6/4)** takes the 6 animal classes as ID classes and 4 vehicle classes as OOD classes. **CIFAR10 (5/5)** has 3 animal and 2 vehicle classes for both ID and OOD classes. We select 400 labeled samples for each ID classes and totally 20,000 unlabeled samples from ID and OOD classes.
- **SVHN (6/4)**: We select the first "0"-"5" as ID classes and the rest as OOD. We select 100 labeled samples for each ID class and totally 20,000 unlabeled samples.
- **CIFAR100 (50/50)**: created from **CIFAR100** (Krizhevsky et al., 2009). The first 50 classes are taken as ID classes and the rest as OOD classes. We select 100 labeled samples for each ID classes and totally 20,000 unlabeled samples.
- **Tiny ImageNet (100/100)**: created from **Tiny ImageNet**, which is a subset of **ImageNet** (Deng et al., 2009) with images downscaled to $64 \times 64$ from 200 classes. The first 100 classes are taken as ID classes and the rest as OOD classes. We select 100 labeled samples for each ID class and totally 40,000 unlabeled samples.

**Here we use C to represent CIFAR, TIN to represent Tiny ImageNet for short**. For each dataset, we use the same experimental setup as in Section 3.

### C.1  IMBALANCE OF PSEUDO-LABELS

To demonstrate the imbalance of pseudo-labels on OOD data, we compute two metrics to measure the extent of imbalance.

- The KL divergence of pseudo-label distribution $q$ and the uniform distribution $u$:

$$kl = KL(q||u) = \sum_i q_i \log \frac{q_i}{u_i}$$

- The ratio of the 'majority class' and 'minority class':

$$r = \frac{\max_i q_i}{\min_i q_i}$$

The results on these datasets are displayed in Table 2.

Table 2: Illustration of the extent of imbalance of pseudo-labels on different datasets.

|       | C10 (6/4) | C10 (5/5) | SVHN (6/4) | C100 (50/50) | TIN (100/100) |
|-------|-----------|-----------|------------|--------------|---------------|
|       |           |           | $kl$       |              |               |
| ID    | 0.0131    | 0.0169    | 0.0006     | 0.0771       | 0.5319        |
| OOD   | 0.3636    | 0.0429    | 0.1031     | 0.4177       | 0.9020        |
|       |           |           | $r$        |              |               |
| ID    | 1.5390    | 1.6235    | 1.1047     | 5.1834       | 107.6684      |
| OOD   | 18.3183   | 2.4878    | 4.4797     | 213.7661     | 1279.9238     |

### C.2  PSEUDO-LABELING STRATEGY FOR OOD DATA

We report the test accuracy of the four pseudo-labeling strategies in Section 3.3. The class-mismatch ratio is set to $100\%$. Table 3 shows the results.

Note that Re-Assigned Labeling has many possibilities. If there are $n$ ID classes and $m$ OOD classes, $A_n^m$ possible assignments exist. It is impossible to experiment on all of them. To deal with it, we randomly choose 10 possible assignment and pick the maximum performance among them.

Table 3: Performance of four different pseudo-labeling strategies on different datasets.

| | C10 (6/4) | C10 (5/5) | SVHN (6/4) | C100 (50/50) | TIN (100/100) |
|---|---|---|---|---|---|
| Baseline | 76.21 (-) | 82.36 (-) | 88.33 (-) | 58.68 (-) | 39.08 (-) |
| Re-Assigned Labeling | 75.99 (-0.22) | 76.90 (-5.46) | 84.40(-3.93) | 50.52(-8.16) | 34.90 (-4.76) |
| Open-Set Labeling | 77.95(+1.74) | 83.54 (+1.18) | 88.31(-0.02) | 58.76(+0.08) | 40.06(+0.86) |
| Oracle Labeling | 79.25(+3.04) | 87.26(+4.9) | 92.07(+3.74) | 63.9(+3.737) | 45.28(+6.78) |

## D    COMPARISON ON MORE DATASETS

We compare our method with vanilla PL and the two class-mismatched methods in Section 6.2. We use the following hyperparameters:

- **CIFAR10 (6/4)**: $\tau = 0.95, \gamma = 0.3, E_{pt} = 50, E_{pl} = 2, K = 4$
- **SVHN (6/4)**: $\tau = 0.95, \gamma = 0.3, E_{pt} = 50, E_{pl} = 2, K = 4$
- **CIFAR100 (50/50)**: $\tau = 0.95, \gamma = 0.18, E_{pt} = 50, E_{pl} = 2, K = 20$
- **Tiny ImageNet (100/100)**: $\tau = 0.9, \gamma = 0.15, E_{pt} = 50, E_{pl} = 2, K = 20$

Table 4: Comparison on four datasets with different class-mismatch ratios. The backbone is Wide-ResNet-28-2 for all experiments.

| Mismatch Ratio | 0% | 25% | 50% | 75% | 100% |
|---|---|---|---|---|---|
| | | | **CIFAR10 (6/4)** | | |
| PL | $78.23 \pm 0.39$ | $76.50 \pm 0.30$ | $75.85 \pm 0.35$ | $74.60 \pm 0.27$ | $73.97 \pm 0.30$ |
| UASD | $78.80 \pm 0.46$ | $77.33 \pm 0.29$ | $77.28 \pm 0.33$ | $76.90 \pm 0.59$ | $76.27 \pm 0.30$ |
| DS$^3$L | $78.21 \pm 0.32$ | $77.83 \pm 0.30$ | $76.78 \pm 0.41$ | $76.28 \pm 0.33$ | $75.27 \pm 0.37$ |
| $\Upsilon$-Model | $\mathbf{78.98 \pm 0.49}$ | $\mathbf{78.93 \pm 0.28}$ | $\mathbf{77.78 \pm 0.31}$ | $\mathbf{77.90 \pm 0.21}$ | $\mathbf{78.50 \pm 0.27}$ |
| baseline | | | $76.21 \pm 0.21$ | | |
| | | | **SVHN (6/4)** | | |
| PL | $92.02 \pm 0.30$ | $90.12 \pm 0.30$ | $88.76 \pm 0.29$ | $87.07 \pm 0.23$ | $86.10 \pm 0.79$ |
| UASD | $\mathbf{92.16 \pm 0.29}$ | $91.32 \pm 0.32$ | $89.34 \pm 0.30$ | $89.26 \pm 0.27$ | $88.62 \pm 0.27$ |
| DS$^3$L | $92.07 \pm 0.33$ | $90.35 \pm 0.30$ | $89.2 \pm 0.56$ | $88.76 \pm 0.27$ | $87.43 \pm 0.31$ |
| $\Upsilon$-Model | $92.15 \pm 0.30$ | $\mathbf{91.48 \pm 0.27}$ | $\mathbf{89.95 \pm 0.30}$ | $\mathbf{89.47 \pm 0.36}$ | $\mathbf{89.58 \pm 0.56}$ |
| baseline | | | $88.33 \pm 0.19$ | | |
| | | | **CIFAR100 (50/50)** | | |
| PL | $61.68 \pm 0.30$ | $60.20 \pm 0.25$ | $60.12 \pm 0.30$ | $57.79 \pm 0.27$ | $57.62 \pm 0.57$ |
| UASD | $60.30 \pm 0.26$ | $59.42 \pm 0.61$ | $59.92 \pm 0.53$ | $58.94 \pm 0.63$ | $58.74 \pm 0.50$ |
| DS$^3$L | $60.68 \pm 0.67$ | $60.60 \pm 0.40$ | $59.22 \pm 0.33$ | $59.74 \pm 0.37$ | $\mathbf{59.56 \pm 0.57}$ |
| $\Upsilon$-Model | $\mathbf{62.10 \pm 0.30}$ | $\mathbf{61.26 \pm 0.40}$ | $\mathbf{60.68 \pm 0.24}$ | $\mathbf{60.12 \pm 0.27}$ | $59.46 \pm 0.36$ |
| baseline | | | $58.68 \pm 0.25$ | | |
| | | | **Tiny ImageNet (100/100)** | | |
| PL | $43.42 \pm 1.03$ | $42.88 \pm 1.51$ | $41.94 \pm 1.49$ | $39.72 \pm 2.30$ | $38.94 \pm 2.41$ |
| UASD | $43.34 \pm 0.78$ | $42.34 \pm 0.61$ | $41.8 \pm 1.338$ | $41.08 \pm 1.16$ | $36.16 \pm 1.05$ |
| DS$^3$L$^*$ | - | - | - | - | - |
| $\Upsilon$-Model | $\mathbf{44.42 \pm 0.43}$ | $\mathbf{43.48 \pm 0.40}$ | $\mathbf{42.42 \pm 0.95}$ | $\mathbf{43.22 \pm 0.60}$ | $\mathbf{41.76 \pm 0.63}$ |
| baseline | | | $39.66 \pm 0.52$ | | |

$^*$ We cannot finish DS$^3$L on a mid-scale dataset like Tiny ImageNet within a reasonable time.

For **CIFAR100 (50/50)** and **Tiny ImageNet (100/100)**, we use a weight factor $\lambda$ to trade off the loss on labeled set $\mathcal{D}_l$ and pseudo-labeled set $\mathcal{P}$, which ramps up with function $\lambda = \exp\left(-5 \times \left(1 - \min\left(\frac{iter}{40,000}, 1\right)\right)^2\right)$, where $iter$ is the number of training steps from $E_{pt}$.

