# OpenReview forum: "On Pseudo-Labeling for Class-Mismatch Semi-Supervised Learning"
_ICLR.cc/2022/Conference — ICLR 2022 Submitted_

### Official Review · Reviewer_kZv3 · 2021-10-30

**Correctness:** 4
**Technical Novelty And Significance:** 4
**Empirical Novelty And Significance:** 3
**Recommendation:** 6
**Confidence:** 3

**Main Review:**

##### Strength

The paper gives insights to the behavior of pseudo-labels under the class-mismatch problem experimentally, and provides an algorithm that can alleviate the issues that are discussed. It performs better than the pseudo-label baseline, but also performs better than two recent class-mismatch SSL baselines (especially with high mismatch ratio), so the benefits shown in the experimental results are one of the main strengths of the paper.

The empirical comparison of open-set labeling and oracle labeling is interesting and motivates the proposed methodology, and makes the story clear. The ablation study further motivates the design of the Upsilon method.

##### Weaknesses

Since Upsilon is based on the findings and discussion in Section 3, it would be better to have at least one more dataset (for example, SVHN that is used in a later section) to see if the findings are not specific to a single dataset.

In Figure 2, it would be interesting to investigate if similar results hold with random splits for ID/OOD classes, instead of the animal/vehicle split. (I think the SVHN experiments later on somehow show that the splits doesn't matter so much, since the 0-5/6-9 split is not as semantically meaningful as the one in CIFAR-10.)

It is interesting and surprising how open-set labeling is worse than oracle labeling, although both are evaluated with only ID classes.  It would be interesting to have some discussions about potential underlying mechanisms of this experimental result.

##### Minor comments
- In the caption of Figure 3, it says "A lot of ID samples are misclassified into one class." but is this correct? Should it be "OOD samples"?
- Which animal class is class 0 in Fig.2(c) and Fig.3(b)?
- Table 1: typo "aof"

=======================
##### After rebuttal

Thank you for the additional experiments and for answering my questions. I would like to keep my positive score. The answer to "It is interesting .. experimental result." makes sense. So there may be some implicit transfer learning going on.

**Summary Of The Paper:**

This paper works on the class-mismatch semi-supervised learning problem, where the assumption is that the unlabeled samples include class labels that do not appear in the labeled data, i.e., out-of-distribution (OOD) data, in addition to the class labels that appear in the labeled data, i.e., in-distribution (ID) data. It is known that if there are OOD data included in the unlabeled data, it can degrade performance of semi-supervised learning algorithms. This paper focuses on one of the semi-supervised learning methods, which is the pseudo-labeling method. The paper first investigates how using pseudo-labels in the class-mismatch semi-supervised learning setup can be problematic. The experiments show that proportions of high-confidence data in ID data is larger than OOD data. Experiments also show that pseudo-labels (w.r.t. class predictions) are more balanced in ID data while it is unbalanced in OOD data at the beginning of training (with a pretrained model). This phenomenon becomes exaggerated at the end of training, where almost all of the OOD data is predicted into a single ID class. Assuming we know the labels of the unlabeled data, the paper performs further experiments that compare a few methods, and finds out that it is ideal if we have access to the underlying single class labels of OOD data, and solve the problem as a multi-class classification problem with K_id + K_ood (number of classes of ID and OOD) classes. Based on these observations, the paper proposes an Upsilon model, that consists of a re-balanced pseudo-labelling (RPL) and semantic exploration clustering (SEC). RPL aims to balance the distributions of pseudo-labels based on the observation that pseudo-labels become balanced/imbalanced with ID/OOD data, respectively. SEC is a workaround for not having ground truth labels of OOD data.  Experiments show comparison with both traditional SSL methods and class-mismatch SSL methods, and show how the proposed Upsilon model works better than these baselines. Ablation study shows why the components within the Upsilon model is necessary.

**Summary Of The Review:**

The paper investigates the issues of using pseudo-labels in class-mismatch semi-supervised learning, and shows the benefits of the proposed method empirically. It performs better than recent class-mismatch semi-supervised learning methods.

---

> ### Author Response · Authors · 2021-11-18
> **Responses to Reviewer kZv3**
>
> We appreciate your precious suggestions. Here are our responses to your questions:
>
> > Since Upsilon .. a single dataset.
>
> > In Figure 2, .. the one in CIFAR-10.)
>
> We answering the two questions altogether here. We experiment on totally 5 datasets CIFAR10 (6/4), CIFAR10-random (5/5), SVHN(6/4), CIFAR100(50/50), Tiny ImageNet (100/100). ($n$/$m$) means $n$ ID classes and $m$ OOD classes. CIFAR10 and SVHN use the same setup as in the paper. CIFAR10-random have $3$ animal and $2$ vehicle classes for both ID and OOD classes. CIFAR100 and Tiny ImageNet split all classes into half, the first half as ID and the rest as OOD. CIFAR100 and Tiny ImageNet are nearly randomly splited for their classes are indexed in alphabetical order. We prove two things: imbalance of pseudo-labels (Section 3.2) and performance of different labeling strategies (Section 3.3). For the former, we compute the imbalance ratio by the KL divergence of pseudo-label distribution $q$ and the uniform distribution $u$, i.e., $kl = KL(q||u)$ and by $r = \max(q) / \min(q)$. We report the imbalance ratio on both ID data and OOD data. For the latter, we report test accuracy with class-mismatch ratio 100% as in the paper.
>
> **Table 1**. Pseudo-label imbalance ratio of different datasets on ID and OOD data.
>
> |     | CIFAR10 | CIFAR10-random |  SVHN  | CIFAR100 | Tiny Imagenet |
> |:---:|:-------:|:--------------:|:------:|:--------:|:-------------:|
> | $kl (\downarrow)$  |         |                |        |          |               |
> | ID |  0.0131 |     0.0169     | 0.0006 |  0.0771  |     0.5319    |
> | OOD |  0.3636 |     0.0429     | 0.1031 |  0.4177  |     0.902     |
> |  $r (\downarrow)$  |         |                |        |          |               |
> |  ID |  1.539  |     1.6235     | 1.1047 |  5.1834  |    107.6684   |
> | OOD | 18.3183 |     2.4878     | 4.4797 | 213.7661 |   1279.9238   |
>
> It can be seen that pseudo-labels on ID data are much more balanced than on OOD data, which is consistent with Section 3.2.
>
> **Table 2**. Labeling strategy on different datasets.
>
> |     | CIFAR10 | CIFAR10-random | SVHN | CIFAR100 | Tiny Imagenet |
> | --- | --- | --- | --- | --- | --- |
> | Baseline | 76.21 | 82.36 | 88.33 | 58.68 | 39.08 |
> | Re-Assigned Labeling | 75.99 (-0.22) | 76.9 (-5.46)| 84.40(-3.93) | 50.52(-8.16) | 34.9 (-4.76)|
> | Open-Set Labeling | 77.95(+1.74) | 83.54 (+1.18 ) | 88.31(-0.02) | 58.76(+0.08) | 40.06(+0.86) |
> | Oracle Labeling | 79.25(+3.04) | 87.26(+4.9) | 92.07(+3.74) | 63.9(+3.737) | 45.28(+6.78) |
>
> Note that Re-Assigned Labeling has many possibilities. If there are $n$ ID classes and $m$ OOD classes, $A_n^m$ possible assignments exist. We can not experiment on all of them. We just randomly choose and pick the maximum performance. Anyway, the conclusion is not changed: Re-Assigned Labeling < Baseline <= Open-Set Labeling < Oracle Labeling.
>
> * * *
>
> > It is interesting ..  experimental result.
>
> We think the most important reason is that Oracle Labeling utilizes information among OOD classes. In the experiment of figure 4, open-set labeling takes all the vehicle samples as one class, ignoring the fact that they actually come from four classes -- airplane, automobile, ship and truck. The difference among data of these classes may provide useful information that is not covered by labeled data of ID classes, especially when the labeled data is not plenty. For example, learning to distinguish between airplane and truck is helpful for distinguishing between bird and dog. But the model trained by open-set labeling loses such benefit. In contrast, Oracle labeling can help the model capture this information by utilizing the ground truth labels. Consequently, oracle labeling performs better than open-set labeling.
>
> * * *
>
> > In the caption of .. "OOD samples"?
>
> Thanks for pointing out the ambiguity. Class 0-5 are ID classes. We can see from Figure 3(b) that a lot of samples of ID classes are predicted as class 0. Classes of 1-5 have 100 - 200 samples misclassified as class 0. The amount is apparently larger than predicted as other classes. That is what the sentence means. We have changed the statement to avoid confusion.
>
> * * *
>
> > Which animal class is class 0 in Fig.2(c) and Fig.3(b)?
>
> Bird.
>
> * * *
>
> > Table 1: typo "aof"
>
> Thanks. We have corrected it in revision.
>
>
> Hoping our responses have answered your questions :).

---

> ### Author Response · Authors · 2021-11-26
> **Thanks for your positive feedback**
>
> We are very grateful for the positive score. Hoping our work is also useful to you :).

---

### Official Review · Reviewer_ayVg · 2021-11-03

**Correctness:** 3
**Technical Novelty And Significance:** 2
**Empirical Novelty And Significance:** 2
**Recommendation:** 3
**Confidence:** 5

**Main Review:**

Strengths:
1. This paper provides the empirical analysis of the Pseudo-Labeling model for ID and OOD data.
2. This paper proposed a novel two-branched Υ-Model to solve the class miss-match issue in SSL.
3. Experiments on different SSL benchmarks empirically validate the effectiveness of the proposed method.

Weakness:
1. The motivation for imbalanced pseudo-labels is not clear.
2. The technical novelty and depth of the proposed approach are limited.
3. The experiments are not extensive.



**Summary Of The Paper:**

This paper focuses on the problem of semi-supervised learning (SSL) with class miss-match. The authors first empirically analyze Pseudo-Labeling (PL) in class-mismatched SSL and proposed a new method that consists of two components – Re-balanced Pseudo-Labeling (RPL) and Semantic Exploration Clustering (SEC). Experiments show that the proposed method achieves steady improvement over supervised baseline and state-of-the-art performance under all class mismatch ratios on different benchmarks.


**Summary Of The Review:**

This paper is well presented and organized. The proposed ideas are interesting. However, there are several main issues as follows:
1. the motivation is not clear and strong. The authors state that " imbalance of pseudo-labels harms the performance”. It lacks justifications for this statement. Fig 3 (b) shows that imbalanced pseudo-labels of OOD would result in many of data with class 1-5 into class 0. However, if the pseudo-labels of OOD is not imbalanced, it may still force the ID data into the incorrect class.
2. Fig 7 (c) shows that Υ-Model is sensitive to the Number of Extra Classes K. However, this paper does not provide a good strategy to choose K.  In the real case, we don’t know the actual number of OOD classes.

3.  The Υ-Model may not work well in a large-scale dataset since the performance of the proposed SEC technical relies on the deep clustering algorithm. When the dataset becomes more complex or the number of OOD classes is large, then SEC may not work at all. It is better to show some results in large-scale datasets.
4. Can Υ-Model extend to other SSL methods? If not, it could be a limation of the proposed method. It is better to show some empirical results.
5. Since 100% Class Mismatch ratio is an extreme case, it is better to show a 50% ratio result for Fig 7 (c).

6. What is the purpose of Fig 2 (a)?

---

> ### Author Response · Authors · 2021-11-18
> **Responses to Reviewer ayVg [1/2]**
>
> Thank you for spending time and giving useful suggestions. Our responses are as follows:
>
> > the motivation is not clear and .. it may still force the ID data into the incorrect class.
>
> We think it is a misunderstanding of our paper. We say imbalance of pseudo-labels harms the performance. But **we do not say imbalance of pseudo-labels is the only reason for performance degradation**. Investigation of Section 3.3 has shown that assigning ID labels to OOD data may damage the performance even if there is no imbalance problem (Figure 4(b)). **And that is exactly conclusion 3 of Section 3.4**. We agree with you but it has been clarified in our paper.
>
> We emphasize the imbalance of pseudo-labels for two reasons. First, **it is one reason for performance degradation** (Section 3.2). Fig 3 (b) shows the prediction on ID samples are affected by pseudo-labels of OOD data. Almost all the OOD samples are predicted as class 0, which means they have pseudo-labels biased to 0. Trained with such imbalanced labels, the model misclassified many ID samples as class 0. Second, **it can be utilized by our method (Section 4.1) to filter out OOD data**. It is proved by Figure 7(a). There may exist other problems when creating pseudo-labels on ID classes for OOD data. But we can utilize the imbalance property to circumvent them.
> * * *
> > Fig 7 (c) shows that Υ-Model .. know the actual number of OOD classes
>
> $K=0$ and $K=1$ do not use clustering. SEC works when $K \ge 2$. The curve in Fig 7 (c) shows that the worst performance ($K=2$)  only differs with the best performance ($K=4$) within about 1% in absolute accuracy. Also, we note that the x-axis is in a log scale. Increasing K exponentially does not change much. Thus we do not think $\Upsilon$-model is sensitive to K.
>
> $K$ can be set as a hyper-parameter to be tuned. The best choice is the actual class number. This number can be given in advance or estimated, e.g., like [1].
> The estimation of the number of classes is orthogonal to this paper. Also, **whether we can accurately set K is not that important, for the head of extra K classes is not used at test time**. It is only assistance during training.
> * * *
> > The Υ-Model may not .. show some results in large-scale datasets.
>
> Deep clustering has been proven effective on large-scale datasets [2] [3]. For resource restriction, we provide results on CIFAR100(50/50) and Tiny ImageNet(100/100), which contain much more classes. ($n$/$m$) means $n$ ID classes and $m$ OOD classes. We compare our method with vanilla PL, UASD and DS3L as in the paper.
>
> **Table 1.** Performance on CIFAR100 (50/50).
>
> |           |       0%      |  25%  |  50%  |  75%  |  100% |
> |:--------:|:-------------:|:-----:|:-----:|:-----:|:-----:|
> |    PL    |     61.68     |  60.2 | 60.12 | 57.79 | 57.62 |
> |   UASD   |      60.3     | 59.42 | 59.92 | 58.94 | 58.74 |
> |   DS3L   |     60.68     |  60.6 | 59.22 | 59.74 | **59.56** |
> |  $\Upsilon$-Model (ours)  |      **62.1**     | **61.26** | **60.68** | **60.12** | 59.46 |
> | baseline |     58.68     |||||
>
> **Table 2.** Performance on Tiny ImageNet (100/100).
>
> |           |       0%      |  25%  |  50%  |  75%  |  100% |
> |:--------:|:-------------:|:-----:|:-----:|:-----:|:-----:|
> |    PL    |     43.42     | 42.88 | 41.94 | 39.72 | 38.94 |
> |   UASD   |     43.34     | 42.34 |  41.8 | 41.08 | 36.16 |
> |   DS3L$^*$   |    -           |   -    |     -  |     -  |     -  |
> |  $\Upsilon$-Model (ours)  |     **44.42**     | **43.48** | **42.42** | **43.22** | **41.76** |
> | baseline |     39.66     |       |       |       |       |
>
> \* : DS3L is too costly for medium and large datasets that we can not finish within a reasonable time.
>
>
>
>
> * * *
> > Can Υ-Model extend .. some empirical results
>
> Our paper mainly investigates how OOD data influence the classification performance of ID data when using pseudo-labeling. We show that pseudo-labels on OOD data is imbalanced and it will harm the performance on ID data. We also investigate what is the best labeling strategy for OOD data. Our method is a modification of the vanilla pseudo-label method according to the analysis. We admit $\Upsilon$-Model is currently designed for pseudo-labeling method. However, we believe our findings and method can provide useful guidance for other methods in class-mismatch setting.

---

> > ### Author Response · Authors · 2021-11-18
> > **Responses to Reviewer ayVg [2/2]**
> >
> > > Since 100% Class .. a 50% ratio result for Fig 7 (c).
> >
> > We provide result in 25%, 50%, 75% and 100% ratio setting.
> >
> > **Table 3.** Influence of the number of extra classes $K$.
> >
> > |  K   |   25%  |   50%  |   75%  |  100% |
> > |:------:|:------:|:------:|:------:|:-----:|
> > |  0  |  76.92 |  76.8  |  75.48 | 75.38 |
> > |  1  |  77.17 |  77.12 |  76.88 | 77.27 |
> > |  2  | 77.083 |  77.3  | **78.383** |  77.4 |
> > |  4  |  **78.93** |  77.78 |  77.9  |  **78.5** |
> > |  8  | 78.017 | **78.167** | 77.683 | 78.08 |
> > | 16  | 77.272 | 77.617 | 77.367 |  77.8 |
> > | 32  | 77.267 |  77.55 | 77.283 | 77.57 |
> >
> > Again $K=0$ and $K=1$ do not actually use SEC. We don't see evidence that our model is sensitive to $K$.
> >
> > * * *
> > > What is the purpose of Fig 2 (a)?
> >
> > Its purpose is stated in Section 3.2 just under figure 2. It says pre-trained model tends to have high confidence on ID data than OOD data. However, in class-mismatched SSL, the unlabeled data are in much larger quantities. When the class mismatch ratio is large, there are quite a few OOD data with high confidence scores. When the mismatch ratio is large, the negative effect of OOD data will overwhelm the possitive effect brought by ID data. And one of the negative effects is imbalance. Fig 2 (a) also clarifies the motivation of Re-balanced Pseudo-Labeling (in Section 4.1) that by selecting only top-N confident samples for each ID class, we will keep ID data and omit many OOD data since confidence on ID data tends to be higher than OOD data.
> >
> >
> > [1] Learning to discover novel visual categories via deep transfer clustering. In ICCV, 2019.
> >
> > [2] Self-labelling via simultaneous clustering and representation learning. ICLR 2020
> >
> > [3]Unsupervised Learning of Visual Features by Contrasting Cluster Assignments. NeurIPS 2020
> >
> > Hoping our responses make clear our ideas to you :).

---

### Official Review · Reviewer_xUsz · 2021-11-03

**Correctness:** 3
**Technical Novelty And Significance:** 3
**Empirical Novelty And Significance:** 3
**Recommendation:** 6
**Confidence:** 4

**Main Review:**

Pros:

(1) OOD SSL is a very interesting topic, and it is also a realistic setting in real world applications.

(2) The insight that pseudo-labeled data is often class-imbalanced is also interesting and can guide future research in this direction.

(3) Authors conducted comprehensive experiments and ablation studies on some small-scale benchmarks.

(4) This paper is well-written and easy to follow.

Cons:

(1) CIFAR-10 and SVHN are much easier than CIFAR-100, TinyImageNet or ImageNet. To show the effectiveness of this method on large-scale dataset, it may be necessary to conduct some experiments on larger benchmarks, especially when the performance improvements on small-scale benchmarks are relatively marginal (about 1.7~2.8% on CIFAR-10 as in Table 1 and ~3% on SVHN as shown in Figure 6).

(2) The current state-of-the-art method on semi-supervised learning is FixMatch [1], however, the authors only compared against worse-performing methods in SSL like Mean Teacher. Therefore, can we still obtain similar observations and probelms when we are using a better SSL framework like Fixmatch? Will the method still perform better when it is integrated with FixMatch?

(3) The OOD SSL is in fact investigated by a con-current work [2], which was released on arXiv about 10 months ago and was in submission to ICLR 2022 too. The rate of this paper will be not reduced for not comparing [2], however, I am curious about the key differences between this paper and [2]. If the author can provide some comparisons and insights, I would be very grateful.

(4) The idea of simply truncating the number of pseudo-labeled data to the minimum is quite brute force, is there any ablation studies on other popular long-tailed recognition method re-weighting, re-sampling, marginal loss or even multi-expert framework?

(5) Will the code of this method be made public available?

(6) Using the expensive and time-consuming clustering method on large-scale benchmark like ImageNet may be of a big challenge to computational efficiency.

[1] Kihyuk Sohn, David Berthelot, Nicholas Carlini, Zizhao Zhang, Han Zhang, Colin Raffel, Ekin Do- gus Cubuk, Alexey Kurakin, and Chun-Liang Li. Fixmatch: Simplifying semi-supervised learning with consistency and confidence. In NeurIPS, 2020.

[2] Cao, Kaidi, Maria Brbic, and Jure Leskovec. "Open-World Semi-Supervised Learning." arXiv preprint arXiv:2102.03526 (2021).
Harvard

**Summary Of The Paper:**

This paper researched on the semi-supervised learning task when there are unlabeled out of distribution data from other classes. Several interesting issues of class mismatch SSL are studied, including the reasons for the performance degradation of PL in OOD data and how to better pseudo-label OOD data to provide a more balanced semantic distribution. To address above-mentioned problems, Y-model, consisting of two components – Re-balanced Pseudo-Labeling (RPL) and Semantic Exploration Clustering (SEC), are proposed. Authors conducted several experiments to show the effectiveness of the proposed method, such as CIFAR-10 and SVHN.

**Summary Of The Review:**

I like this paper in general, but I have some concerns on the effectiveness of this method in large-scale benchmarks. Also using the expensive and time-consuming clustering method on large-scale benchmark like ImageNet may be of a big challenge to computational efficiency.

---

> ### Author Response · Authors · 2021-11-18
> **Responses to Reviewer xUsz [1/2]**
>
> We are very grateful for your suggestions. Our responses are as follows:
> >(1)	CIFAR-10 and SVHN are much easier .. on SVHN as shown in Figure 6).
>
> Due to hardware resource restrictions, we can not conduct experiments on large-scale datasets like ImageNet. Sorry for that. However, we add experiments on CIFAR100(50/50) and Tiny ImageNet(100/100). ($n$/$m$) means $n$ ID classes and $m$ OOD classes. For each dataset, we select first half of classes as ID classes, with 100 labeled samples for each ID class. The rest half of the classes are OOD classes. For CIFAR100, 20000 unlabeled data are selected according to different class-mismatch ratios. For Tiny ImageNet, there are 40000 unlabeled data.  We compare our method with vanilla Pseudo-Labeling(PL), D3SL and UASD (as in the paper) with varied class-mismatch ratios (0%,25%,50%,75%,100%).
>
> **Table 1.** Performance on CIFAR100 (50/50).
>
> |           |       0%      |  25%  |  50%  |  75%  |  100% |
> |:--------:|:-------------:|:-----:|:-----:|:-----:|:-----:|
> |    PL    |     61.68     |  60.2 | 60.12 | 57.79 | 57.62 |
> |   UASD   |      60.3     | 59.42 | 59.92 | 58.94 | 58.74 |
> |   DS3L   |     60.68     |  60.6 | 59.22 | 59.74 | **59.56** |
> |  $\Upsilon$-Model (ours)  |      **62.1**     | **61.26** | **60.68** | **60.12** | 59.46 |
> | baseline |     58.68     |||||
>
> **Table 2.** Performance on Tiny ImageNet (100/100).
>
> |           |       0%      |  25%  |  50%  |  75%  |  100% |
> |:--------:|:-------------:|:-----:|:-----:|:-----:|:-----:|
> |    PL    |     43.42     | 42.88 | 41.94 | 39.72 | 38.94 |
> |   UASD   |     43.34     | 42.34 |  41.8 | 41.08 | 36.16 |
> |   DS3L$^*$   |    -           |   -    |     -  |     -  |     -  |
> |  $\Upsilon$-Model (ours)  |     **44.42**     | **43.48** | **42.42** | **43.22** | **41.76** |
> | baseline |     39.66     |       |       |       |       |
>
> \* : DS3L is too costly for medium and large datasets that we can not finish it within a reasonable time.
>
> The improvement seems marginal, but we want to make clear that most of the traditional SSL methods will underperform supervised baseline when the proportion of OOD data (usually > 50%). The OOD data is usually difficult to exploit since they come from a totally different distribution. Using these data also puts the model at risk of performance degradation. Thus at large mismatch ratio, how to avoid performance degradation is the primary mission. Our method can further improve the performance, which is very precious.
> * * *
> >The current state-of-the-art .. method still perform better when it is integrated with FixMatch?
>
> - We want to clarify that the reason why we do not compare with FixMatch is that **augmentation plays an important role in FixMatch**. Our paper investigates how the created pseudo-labels influence performance. **Augmentation is orthogonal to our investigation**. So we avoid the influence brought by different augmentation, since labeled data can also be used as unlabeled data (this trick is also used in FixMatch). In all our compared methods, as well as supervised baseline and our method, we use the same augmentation on both labeled and unlabeled data. However, we still display the results of FixMatch here. Experiments are conducted on CIFAR10 with the same setup in the paper. 'paper' means the augmentation used in the paper which is commonly used in class-mismatch setting [3] [4].
>
> **Table 3.** FixMatch results with different augmentation.
>
> |          | Augmentation |   0   |   25%  |   50%   |   75%   |   100%  |
> |:--------:|:------------:|:-----:|:------:|:-------:|:-------:|:-------:|
> |    PL    |     paper    | 78.23 |  76.5  |  75.85  |   74.6  |  73.97  |
> | FixMatch |     paper    | 78.75 |  76.25 | diverge | diverge | diverge |
> | $\Upsilon$-Model (ours)  |     paper    | **78.98** |  **78.93** |  **77.78**  |   **77.9**  |   **78.5**  |
> | Baseline |     paper    | 76.21 |||||
> |    PL    |    RandAug   |  88.3 |  86.8  |  85.61  |  83.72  |  81.62  |
> | FixMatch |    RandAug   | 87.35 | 84.314 |  82.367 |  78.733 | diverge |
> |   $\Upsilon$-Model (ours)  |    RandAug   | **88.35** |  **86.83** |  **85.91**  |  **85.25**  |  **84.31**  |
> | Baseline |    RandAug   | 83.75 |||||
>
> - FixMatch suffers from performance degradation like other SSL methods. Also, FixMatch presents a more serious imbalanced problem than vanilla pseudo-labeling, which makes it unstable during training. We think it is attributed to the fact that  FixMatch creates pseudo-labels by a weak augmented view.
> - Different from FixMatch, our method is an 'offline version' of pseudo-labeling. We update pseudo-labels when passing the whole unlabeled data. For stability, we use no augmentation (also can be seen as the weakest augmentation) when creating pseudo-labels. For fairness, we use the same augmentation as labeled data. However, the table above also shows that applying strong augmentation can improve our method. Therefore, integrated with FixMatch's augmentation strategy, it will improve.

---

> > ### Author Response · Authors · 2021-11-18
> > **Responses to Reviewer xUsz [2/2]**
> >
> > > The OOD SSL is .., I would be very grateful.
> >
> > Thanks for reminding us of this paper. [2] is a good paper and has a lot of useful insights. It differs from our paper in the following points:
> > 1. The main difference is the setting. [2] solves the new open-world semi-supervised learning setting. Similar to our class-mismatched setting, this setting is also given a labeled set with seen classes and an unlabeled set with possibly unseen classes. However, in addition to accurately recognizing previously seen classes, it is required to discover novel classes. Open-world is an extensive setting of ours.
> > 2. Due to differences in settings, the main focus is different. Our paper focuses on investigating how the OOD data influence the performance of classifying ID data (Section 3.2) and how to effectively use them (Section 3.3), while [2] focuses on how to practically solve the newly proposed problem.
> > 3. It seems that our method can be extended to the open-world setting since we also discover novel classes by the clustering method. However, we want to emphasize that our use of clustering is based on the observation that training OOD data with their ground truth benefits the classification of ID data (conclusion of Section 3.3), even though we do not need to classify OOD data. [2] does not investigate that.
> > * * *
> > > The idea of simply .. even multi-expert framework?
> >
> > We want to emphasize that truncating the number of pseudo-labeled data is **not just for solving the imbalance problem brought by OOD data, but also we utilize this property to filter out OOD data with mistakenly high confidence**. Section 4.1 have explained that. By using truncating, a lot of OOD data will be filtered out since they tend to have lower confidence than ID data. With RPL, the high-confidence OOD data will become less and less (Figure 7(a)). However, long-tailed recognition methods do not have this functionality. Anyway, we compare our method with a re-weighting strategy used in ReMixMatch[3], which creates the weight by the ratio of predicted label distribution to real label distribution. We name it PL-reweight. We give a comparison with vanilla PL, RPL and the full $\Upsilon$-Model.
> >
> > **Table 4.** Comparison of reweight strategy and our method.
> >
> > |             |    0   |   25%  |   50%  |   75%  |   100%  |
> > |:-----------:|:------:|:------:|:------:|:------:|:-------:|
> > |      PL     |  78.23 |  76.5  |  75.85 |  74.6  |  73.97  |
> > | PL-reweight | 78.483 | 76.983 | 76.333 | 74.517 | diverge |
> > |     RPL (ours)    |  78.58 |  76.92 |  76.8  |  75.48 |  75.38  |
> > |  $\Upsilon$-Model (ours)   |  78.98 |  78.93 |  77.78 |  77.9  |   78.5  |
> > |   Baseline  |  76.21 |||||
> >
> > The reweighting strategy may cause unstability for the pseudo-labels are changing every time. When the class-mismatch ratio is large, the weight vary on a large scale since the pseudo-labels becomes very imbalanced. We can see reweight underperform RPL and can not work on 100% setting.
> > * * *
> > > Will the code of this method be made public available?
> >
> > Yes, we will soon release the code. It is in a mess now and we are working on polishing it.
> >
> > * * *
> > > Using the expensive .. big challenge to computational efficiency.
> >
> > Yes, it is the case. But actually, it is not necessary to do very well on clustering. The classifier on extra classes is not used during test time. It can be seen as assistance during training. Figure 7(c) has shown the effect of $K$, the number of clusters. When $K=1$, we do not perform clustering but it still improves the performance. However, if you want to further improve the performance, you need to perform better clustering at cost of time. It is actually a trade of time and performance.
> >
> > [1] Fixmatch: Simplifying semi-supervised learning with consistency and confidence. In NeurIPS, 2020.
> >
> > [2] "Open-World Semi-Supervised Learning." arXiv preprint arXiv:2102.03526 (2021)
> >
> > [3] Realistic evaluation of deep semi-supervised learning algorithms. In NeurIPS, 2018.
> >
> > [4] Safe deep semi-supervised learning for unseen-class unlabeled data. In ICML, 2020.
> >
> >
> > Hoping our responses have answered your questions :).

---

### Official Review · Reviewer_A13X · 2021-11-03

**Correctness:** 4
**Technical Novelty And Significance:** 4
**Empirical Novelty And Significance:** 3
**Recommendation:** 6
**Confidence:** 4

**Main Review:**

Strength

- This paper has an interesting observation on how does OOD data influence pseudo-label and resulting model performance.
- The proposed method also seems to be original.

Weakness

I have several concerns about the experiments.
1. First, as the vanilla semi-supervised learning baselines, stronger methods should be leveraged. Especially, as the problem definition of this paper is related to the pseudo-label in semi-supervised learning, the recent “pseudo-label based” semi-supervised learning methods such as FixMatch [1] and ReMixMatch [2] should be considered. VAT or Mean Teacher seems to be outdated. It will also be interesting to investigate if we combine the proposed r-Model with the existing pseudo-labeling-based semi-supervised learning methods.

[1] Sohn et al., Fixmatch: Simplifying semi-supervised learning with consistency and confidence. NeurIPS 2020.

[2] ReMixMatch: Semi-Supervised Learning with Distribution Matching and Augmentation Anchoring. ICLR 2020.

2. It would be better to evaluate the proposed method on other datasets such as CIFAR100, STL, or maybe ImageNet dataset. In order to comprehensively understand the behavior of the proposed method, the proposed method should be evaluated on other datasets (at least on the MNIST dataset).


**Summary Of The Paper:**

The authors tackle the pseudo-label in class-mismatched semi-supervised learning when there are unlabeled out-of-scope data from other classes. The authors propose a model consisting of (1) Re-balanced Pseudo-Labeling, which re-balances pseudo-labels on ID classes to filter out OOD data, and (2) Semantic Exploration Clustering, which uses balanced clustering on OOD data to create pseudo-labels on extra classes.

**Summary Of The Review:**

This paper has an interesting observation and the proposed method seems to be original. However, as mentioned before, it would have been better to have additional training setups and baselines to emphasize the efficacy of the proposed method. I would like to rate this paper to be borderline, but I am open to changing the score after the rebuttal.

=======================

After rebuttal:

After reading the authors' response, I think my concerns both on the compared baselines (FixMatch) and the evaluated datasets (ImageNet) are somewhat addressed. Therefore, I would like to increase the score to be 6.

---

> ### Author Response · Authors · 2021-11-18
> **Responses to Reviewer A13X**
>
> Thanks for your precious suggestions. Our responses are as follows:
>
> > 1.  First, as the vanilla semi-supervised .. semi-supervised learning methods.
>
> We conducted experiments on FixMatch\[1\] and MixMatch\[3\]. Note that we display the result of MixMatch instead of ReMixMatch\[2\] for the reason that MixMatch is simpler and ReMixMatch adds the rotation loss which is orthogonal to this paper. Before we go ahead to the results, there is one thing we want to emphasize: **augmentation plays an important role in these methods, which is orthogonal to this paper**. To be specific, FixMatch uses a much stronger augmentation than labeled data and the anchor view. MixMatch uses mixup operation to create synthesized samples. Our paper investigates how the created pseudo-labels influence performance. So to get rid of other factors (like augmentation) that may affect the performance, **we treat all labeled and unlabeled data equally by applying the same augmentation and ignore other tricks like mixup** in our paper. But here, we display the results of these methods. Note that for a fair comparison, we also use same augmentation for labeled and unlabeled data here. The augmentation 'paper' means what we used in the paper. It is commonly used in class-mismatched settings [4] [5]. RandAug [6] is the augmentation used in FixMatch.
>
> **Table 1.** Results of FixMatch, MixMatch, Pseudo-Labeling (PL) and our method.
>
> |         |  Augmentation |     0%   |    25%  |    50%   |   75%   |   100%  |
> |:--------:|:------------:|:------:|:------:|:-------:|:-------:|:-------:|
> |    PL    |     paper    | 78.23 |  76.5  |  75.85  |   74.6  |  73.97  |
> | MixMatch |     paper    | 77.88 |  77.15 |  76.83 |   75.9  |  75.85  |
> | FixMatch |     paper    |  78.75 |  76.25 | diverge | diverge | diverge |
> |  $\Upsilon$-Model (ours)  |     paper    |  **78.98** |  **78.93** |  **77.78**  |   **77.9**  |   **78.5**  |
> | Baseline |     paper    |  76.21 |||||
> |   PL    |    RandAug   |  88.3 |  86.8  |  85.61  |  83.72  |  81.62  |
> | MixMatch |    RandAug   | 81.98 | 80.08 | 76.75   |  73.45  |  69.33 |
> | FixMatch |    RandAug   |  87.35 | 84.31 |  82.37 |  78.73 | diverge |
> |   $\Upsilon$-Model (ours)  |    RandAug   |  **88.35** |  **86.83** |  **85.91**  |  **85.25**  |  **84.31**  |
> | Baseline |    RandAug   |  83.75 |||||
>
> There are some primary conclusions:
> - FixMatch is unstable in class-mismatched setting while MixMatch is more stable. The former is mainly caused by the imbalance of pseudo-labels. MixMatch's stability may be brought by its mixup operation.
> - FixMatch and MixMatch all suffer from performance degradation in such a setting.
> - The strong augmentation strategy in FixMatch can bring improvement to our method. It brings improvement to all the methods, but we also emphasize that it also pulls up the baseline.
>
>
> * * *
> >2.	It would be better to evaluate the proposed .. on other datasets (at least on the MNIST dataset).
>
> STL10 does not provide labels of its large portion of unlabeled data so it is impossible to create class-mismatched dataset. Also, MNIST is too similar to SVHN. Therefore, we decide to add additional experiments on CIFAR100(50/50) and Tiny ImageNet(100/100). ($n$/$m$) means $n$ ID classes and $m$ OOD classes. For each dataset, we select the first half of classes as ID classes, with 100 labeled samples for each ID class. The rest half of the classes are OOD classes. For CIFAR100, 20000 unlabeled data are selected according to different class-mismatch ratio (0%,25%,50%,75%,100%). For Tiny ImageNet, there are 40000 unlabeled data.  We report the comparison with vanilla PL and two class-mismatched methods D3SL and UASD (as in the paper) here.
>
> **Table 1.** Performance on CIFAR100 (50/50).
>
> |           |       0%      |  25%  |  50%  |  75%  |  100% |
> |:--------:|:-------------:|:-----:|:-----:|:-----:|:-----:|
> |    PL    |     61.68     |  60.2 | 60.12 | 57.79 | 57.62 |
> |   UASD   |      60.3     | 59.42 | 59.92 | 58.94 | 58.74 |
> |   DS3L   |     60.68     |  60.6 | 59.22 | 59.74 | **59.56** |
> |  $\Upsilon$-Model (ours)  |      **62.1**     | **61.26** | **60.68** | **60.12** | 59.46 |
> | baseline |     58.68     |||||
>
> **Table 2.** Performance on Tiny ImageNet (100/100).
>
> |           |       0%      |  25%  |  50%  |  75%  |  100% |
> |:--------:|:-------------:|:-----:|:-----:|:-----:|:-----:|
> |    PL    |     43.42     | 42.88 | 41.94 | 39.72 | 38.94 |
> |   UASD   |     43.34     | 42.34 |  41.8 | 41.08 | 36.16 |
> |   DS3L$^*$   |    -           |   -    |     -  |     -  |     -  |
> |  $\Upsilon$-Model (ours)  |     **44.42**     | **43.48** | **42.42** | **43.22** | **41.76** |
> | baseline |     39.66     |       |       |       |       |
>
> \* : DS3L is too costly for medium and large datasets that we can not finish within a reasonable time.
>
> Our method improves PL steadily and surpasses two related methods. We hope these results solve your concerns:).

---

> > ### Author Response · Authors · 2021-11-18
> > **References**
> >
> > [1] Fixmatch: Simplifying semi-supervised learning with consistency and confidence. NeurIPS 2020.
> >
> > [2] ReMixMatch: Semi-Supervised Learning with Distribution Matching and Augmentation Anchoring. ICLR 2020.
> >
> > [3] MixMatch: A Holistic Approach to Semi-Supervised Learning. NeurIPS 2019
> >
> > [4] Realistic evaluation of deep semi-supervised learning algorithms. In NeurIPS, 2018.
> >
> > [5] Safe deep semi-supervised learning for unseen-class unlabeled data. In ICML, 2020.
> >
> > [6] RandAugment: Practical Automated Data Augmentation with a Reduced Search Space. NeurIPS 2020

---

> ### Author Response · Authors · 2021-11-26
> **Thanks for raising the score**
>
> We are very thankful for your positive feedback! We hope our work is also helpful to you:).

---

### Author Response · Authors · 2021-11-22
**General Responses**

We would like to thank all the reviewers for spending time reviewing and providing valuable suggestions. We have uploaded a revised version of the paper. Here, we want to answer some commonly asked questions.

**Q1. Why don't we compare other state-of-the-art pseudo-label methods like FixMatch and MixMatch?**

Augmentation plays an important role in these methods, which is orthogonal to this paper. For example, FixMatch uses a much stronger augmentation than labeled data and the anchor view. MixMatch uses mixup operation to create synthesized samples. Our paper investigates how the created pseudo-labels influence performance. Therefore, to get rid of other factors (like augmentation) that may affect the performance, we just apply the same augmentation for both labeled and unlabeled data in all the methods and in all the experiments. In this circumstance, comparing these methods is not fair. That is why we do not include these methods in our paper.

**Q2. The universality of analysis conclusions and the efficacy of our method on more datasets.**

We have updated the appendix mainly for this question.

**Q3. Is the clustering method too time-consuming?**

It depends on whether you want to spend more time training to improve performance. SEC aims to simulate the process of Oracle Labeling in Section 3.3, which is the best labeling strategy for OOD data. However, even if we do not perform clustering and just label the OOD data as a unified class, like Open-Set Labeling, it still improves. The more you want to improve, to better clustering you need to perform, at the cost of time. It is actually a trade-off between time and performance.

Again, we want to thank all the reviewers for their precious suggestions.

Sincerely,

The authors

---

### Decision · Program_Chairs · 2022-01-20

**Decision:**

Reject

**Comment:**

All reviewers agreed that this work on OOD and pseudo-labeling presents interesting and strong results. The authors’ rebuttal has addressed some of reviewers’ concerns. Based on the current review and discussion, there are still several major concerns towards the expensive computational cost introduced by the clustering method, the lack of discussion around how the proposed work can be incorporated into SSL methods, and the sensitivity towards the selection of K.